# An Analysis of the Relationship between Energy Trilemma and Economic Growth

**Hyunsoo Kang**

Department of International Trade, Wonkwang University, Iksan 54538, Korea; agkang75@wku.ac.kr

**Abstract:** This study analyzed the relationship between energy trilemma (ET) and economic growth in 109 countries between 2000 and 2020 across income levels and regions. This study constructed an extended Cobb-Douglas production function including three elements of ET such as energy security, energy equity, and environmental sustainability as their effects on economic growth differ by income level and region. The methodology of this study differs from that of previous studies, which utilized the representative value of ET based on principal component analysis. To analyze the panel series, this study utilized econometric procedures, panel regression of pooled ordinary least squares (OLS), feasible generalized least squares (FGLS), fixed effects, and dynamic panel analysis of generalized methods of moments (GMM) by three income levels. In addition, this study undertook a time series analysis between ET and economic growth for each country. The results showed that each element of ET is not balanced. Moreover, each element can contribute differently to economic growth due to differences in income levels and regions. This study suggested that a balanced environmental policy reflecting various aspects of ET is required and can contribute to the economic growth.

**Keywords:** energy trilemma; economic growth; energy security; energy equity; environmental sustainability



## 1. Introduction

The International Energy Agency (IEA) expects that the demand for power, heat, and transport will grow as the world population increases. Therefore, global energy demand will increase rapidly and even potentially double by 2050 [1]. To satisfy this demand, the investment requirements for energy infrastructure must be met. However, we simultaneously face the three challenges of energy investment, energy demand, and supply. Importantly, tackling these challenges implies a trade-off in choosing one over the other. Thus, we call these three elements the energy trilemma (ET).

ET is the energy system's performance across three dimensions, including energy security, energy equity, and environmental sustainability [2]. According to the World Energy Council (WEC), energy security refers to a country's capacity to meet its current and future energy demand reliably and the resilience of its energy infrastructure. Energy equity is a country's ability to provide reliable, affordable, and abundant energy for domestic and commercial use. Environmental sustainability measures the transition of a country's energy system toward low- or zero-carbon by reducing carbon dioxide ($CO_2$) emissions. WEC's world ET index insights into a country's relative energy performance regarding these three dimensions and the accessibility of energy policies for enabling balanced transition management.

Ekins [3] emphasized that the nature of the relationship between economic growth and energy security (including environmental sustainability) has been controversial for a long time. For example, Le and Nguyen [4] argued that energy security seemingly enhances economic growth, while energy insecurity negatively affects economic growth. That is, maintaining excess energy capacity over demand is critical to continuous economic growth. Notably, today's global economy has become less energy intensive by using fewer fossil

fuels for economic growth [4]. By contrast, Demaria [5] mentioned that environmental sustainability and a country's economic growth are not compatible; the amount of energy and material usage increases with economic growth, and consequently, environmental quality will deteriorate.

Despite their interrelationships, energy security, energy equity, and environmental sustainability have been pursued as separate themes, even though all of these elements of ET are important factors affecting economic growth [4]. For example, Khan et al. [6] mentioned that the impact of ET on economic growth is significant only in the long run, especially in developing countries that face the dilemma of trade-off between economic growth and environmental protection (or energy security). Therefore, these countries prefer economic growth rather than a sustainable environment and maintaining energy security. Similarly, Grigoryev and Medzhidova [7] described the "global energy trilemma": economic growth has a higher priority than environmental protection in developing countries where poverty is higher. The authors argued that these countries suffer from an ET or a trade-off between economic growth and environmental sustainability.

The main objective of this study was to investigate the effects of the ET on economic growth across income levels and regions. To identify the relationship between ET and economic growth, this study constructed an extended Cobb-Douglas production function, including the three ET dimensions, and adopted several types of econometric procedures, including panel data analysis by pooled ordinary least squares (OLS) and feasible generalized least squares (FGLS), dynamic panel analysis by difference and system generalized method of moments (GMM), and time series analysis of the simple relationship between one dimension of ET and economic growth by region.

The remainder of this study is organized as follows. Section 2 introduces the previous literature on the relationship between ET and economic growth. Section 3 presents the economic model's specifications and describes the data. Finally, Sections 4 and 5 note the empirical results and conclusions, respectively.

## 2. Literature Review

The WEC stated that any government needs to find the optimal answer to ensure: (1) the availability of energy in sufficient quantities and at reasonable prices; (2) the reliability and safety of energy supply; and (3) its environment friendliness. Energy is the most important input of economic growth since almost all production and consumption activities require energy. Historically, energy has always been at the center of economic growth; more recently, it has been strongly associated with economic growth [6,8]. Another important factor of economic growth are the changes in the government's energy-related policies that affect energy supply, demand, and price [9].

The traditional economic growth theory of "technology-enhanced labor productivity" focuses on economic growth by accumulating capital and knowledge stocks. However, except for changing production factors, this basic growth theory cannot explain the factors responsible for economic depression. For example, it does not consider energy utilization and price volatility [10].

Many studies have investigated the relationship between ET and economic growth from each of the three dimensions: energy security, energy equity, and environmental sustainability on economic growth. The first theme on ET's relationship with economic growth is the relationship between energy security and economic growth. Examining trade with other countries, Gasparatos and Gadda [11] found that an increase in imports from developing countries and tariff barriers is associated with access to resources and affects long-term economic sustainability in Japan. Others notes that energy consumption positively affects economic growth if the government prioritizes a low-energy consumption regime. Moreover, energy consumption policy-related energy security affects economic growth [12]. However, Ozturk et al. [13] suggested that energy consumption positively affects economic growth if the benefit exceeds the externality of energy use. However, the authors find no evidence of causality between energy security and economic growth across

the three income groups. Similarly, Szustak et al. [14] showed that energy production does not directly affect GDP but influences climate change and sustainable development in the long run.

The second theme is the relationship between energy equity and economic growth. Ullah et al. [15] investigated a comprehensive index of energy equity based on four dimensions: energy service availability, clean energy, energy governance, and energy affordability. The authors found a significant relationship between energy poverty and economic growth in both the short and long run, even if the energy supply still faces threats to environmental quality. Simultaneously, countries that require improved energy access and affordability also need suitable energy investment policies to support economic growth [16]. Ziolo et al. [17] also found that economic growth increases both energy efficiency and greenhouse gases. However, if efficiency improves due to consistent financial support, emissions reductions and sustainable economic development can be achieved. Chien and Hu [18] argued that among the various kinds of energy sources, increasing renewable energy capacity has a significant positive influence on macroeconomic efficiency and capital formation. Therefore, there is a positive relationship between renewable energy and economic growth. However, this relationship between renewable energy and economic growth can depend on the country's economic situation and income level [19].

The last theme is the relationship between environmental sustainability and economic growth. Ekins and Jacobs [20] mentioned that environmental sustainability and economic growth are not compatible: economic activities principally lead to land degradation and, consequently, can contribute to deforestation and overgrazing due to agriculture. Drews et al. [21] highlighted that environmental protection and sustainability typically tend to be prioritized over economic growth even if there is a belief of compatibility between environmental sustainability and economic growth. By contrast, Stjepanović et al. [22] noted that both economic growth and environmental sustainability are the main factors for sustaining growth and development, and that we need a new paradigm called the "green growth" approach. That is, green growth ensures that natural assets continue to provide resources and environmental services while we promote economic growth and development.

Based on the literature, this study attempts to analyze the relationship between ET and economic growth by income level. Some studies are similar to the main concept of this study. For example, based on principle components analysis (PCA), Khan et al. [6] utilized the aggregated index for the three dimensions of ET and analyzed the impact of ET on economic growth. Similarly, Fu et al. [22] investigated the relationship between energy security, energy equity, and environmental sustainability from the perspective of economic growth and $CO_2$ emissions. Above all, the PCA approach, unlike conventional methods, has the important advantage of excluding ad-hoc and subjective weights to different indicators. Therefore, it is used for statistical data analysis, data compression, and data noise removal [23]. However, PCA is mixed with various individual characteristics of the original data, and it is difficult to explain the meanings of the three dimensions of ET [24]. In addition, following previous studies, this study emphasizes that all three dimensions of ET are crucial factors affecting economic growth.

Here, this study analyzed the relationship between ET and economic growth for three income groups and regions using panel and time-series data together. This study does not use the PCA approach of previous studies. Rather, in this study, an investigation is possible through the specified models containing the characteristics of each ET index.

## 3. Methodology and Data

### 3.1. World Energy Trilemma Index (WETI) and Current Situation

WEC's world energy trilemma index (WETI), published in 2010, is one of the most comprehensive and informative on countries' energy performance in terms of energy security, energy equity, and environmental sustainability [25]. Although the definition of ET is slightly different among organizations, the fundamental ET is made up of economic

development, energy security, and environmental concerns [2]. Figure 1 shows that sustainable energy development (the overlapping area in the Venn diagram) requires all three interconnected priorities, while the remaining areas indicate the trade-off between the two dimensions of ET.

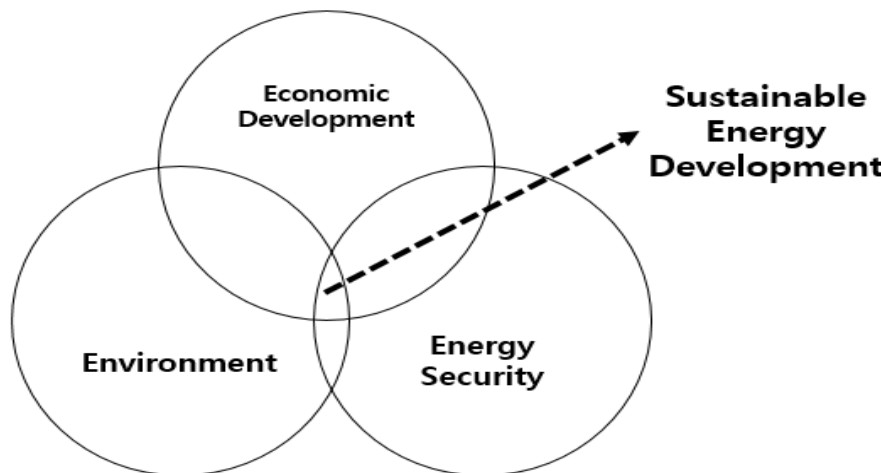

**Figure 1.** The energy trilemma. Source: Reprinted from [2].

Table 1 lists WETI's structure. This index is the total sum of energy security (30%), energy equity (30%), and environmental sustainability (30%), with the remaining reflecting the country's context (10%), including macroeconomic circumstances, management methods, investment stability, and innovations. WETI reflects a nation's capacity for energy demand and supply, including accessibility and affordability for energy use, and energy system's transition. Therefore, the WETI is very useful for suggesting a policy that enables balanced energy transition management, and for performing an analysis that uses the experiences of countries with relevant socioeconomics and energy infrastructure [25]. For example, Barnes and Floor [26] suggested that the energy problems of the developing countries are both serious and widespread due to the lack of access to sufficient and sustainable energy supplies. Therefore, energy and environmental policies are built at the national and international levels.

Table 2 lists the top WETI performers in 2020. The top five overall performing countries are mostly in Europe, whereas the top five improving countries are less developed countries in Africa and Asia. The top three performers of energy security are Canada, Finland, and Romania. According to the IEA [1], Canada has high energy security owing to its natural resource endowment. Finland is a strong leader in the Nordic area and focuses on decarbonizing its energy systems, while Romania benefits from a hydrocarbon oil producer. In terms of energy equity, Luxembourg ranks the first in energy equity aided by its wealth and size, and being a country with extensive natural resources for energy generation [1]. Similarly, the other top four energy equity performers are nations with rich natural energy resources, which helps keep energy prices and affordable. The top three performers of environmental sustainability are Switzerland, Sweden, and Norway. They have maintained their top positions since the middle of 2000, signifying the importance of long-term planning of low-carbon futures [1].

Figure 2 displays the average WETI performance during 2000–2020 according to income level. Notably, the average change ratio of energy equity for low and lower-middle income countries is the largest. In particular, the IEA [1] noted that the major improvers (especially, Mozambique, Cambodia, and Ethiopia) are sub-Saharan African countries which have made significant strides with energy access and affordability. Meanwhile, most developed countries already have 100% energy access. Therefore, affordability is the key differentiator in the energy equity performance of these two sets of countries.

**Table 1.** The structure of WEC's world energy trilemma index (WETI).

| Dimension | Definition | Indicator Category | Indicator |
|---|---|---|---|
| Energy security | Reflects a nation's capacity to meet current and future energy demand reliably, withstand and bounce back swiftly from system shocks with minimal disruption to supplies | A1: Security of supply and energy demand | a: Diversity of primary energy supply |
| | | | b: Import dependence |
| | | A2: Resilience of energy systems | a: Diversity of electricity generation |
| | | | b: Energy storage |
| | | | c: System stability and recovery capacity |
| Energy equity | Assesses a country's ability to provide universal access to affordable, fairly priced and abundant energy for domestic and commercial use | B1: Energy access | a: access to electricity |
| | | | b: access to clean cooking |
| | | B2: Quality energy access | a: access to "modern" energy |
| | | B3: Affordability | a: Electricity prices |
| | | | b: Gasoline and diesel prices |
| | | | c: Natural gas prices |
| | | | d: Affordability of electricity for residents |
| Environmental sustainability | Represents the transition of a country's energy system towards mitigating and avoiding potential environmental harm and climate change impacts | C1: Energy resource productivity | a: Final energy intensity |
| | | | b: Efficiency of power generation and T&D |
| | | C2: Decarbonization | a: Low carbon electricity generation |
| | | | b: GHG emissions trend |
| | | C3: Emissions and pollution | a: $CO_2$ intensity |
| | | | b: $CO_2$ emissions per capita |
| | | | c: CH4 emissions per capita |
| | | | d: PM2.5 mean annual exposure |
| | | | e: PM10 mean annual exposure |

Source: Reprinted from [1].

**Table 2.** 2020 top WETI performers.

| | Top five overall performers and improvers | | | |
|---|---|---|---|---|
| **Rank** | **Performers** | | **Improvers** | |
| | **Country** | **Score** | **Country** | **Improvement since 2000** |
| 1 | Switzerland | 84.3 | Cambodia | 77% |
| 2 | Sweden | 84.2 | Myanmar | 50% |
| 3 | Denmark | 84.0 | Kenya | 41% |
| 4 | Austria | 82.1 | Bangladesh | 38% |
| 5 | Finland | 82.1 | Honduras | 36% |
| | **Top five energy trilemma performers** | | | | |
| **Rank** | **Energy Security** | | **Energy Equity** | | **Environmental Sustainability** | |
| | **Country** | **Score** | **Country** | **Score** | **Country** | **Score** |
| 1 | Canada | 77.1 | Luxembourg | 99.9 | Switzerland | 90.0 |
| 2 | Finland | 75.4 | Qatar | 99.8 | Sweden | 87.5 |
| 3 | Romania | 74.5 | Kuwait | 99.8 | Norway | 87.2 |
| 4 | Denmark | 74.4 | UAE | 99.8 | Albania | 85.8 |
| 5 | Latvia | 74.1 | Oman | 99.7 | France | 85.5 |

Source: Reprinted from [1].

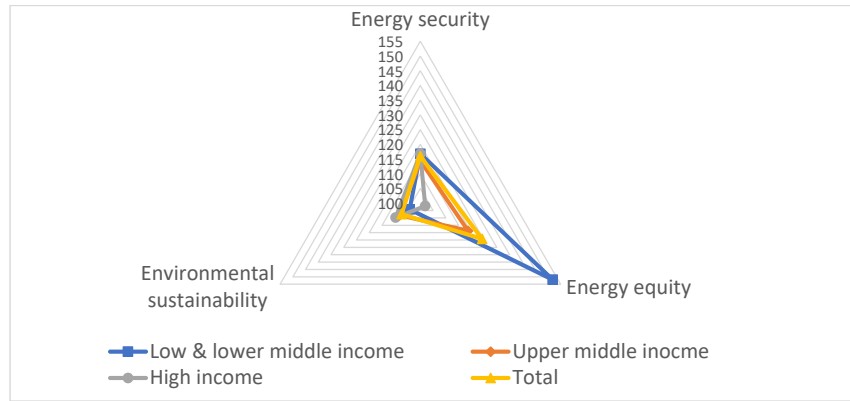

**Figure 2.** Average energy trilemma performance during 2000–2020 by income levels.

Figure 3 shows the simple relationship between the three dimensions of ET and economic growth for the full sample. The slope of the logarithmic regression line implies that a 1% increase in GDP is associated with X% increase in energy security (ES), energy equity (EE), and environmental sustainability (ESUS) performance, respectively. Since Figure 3 does not consider other factors that affect economic growth, the results can only identify the simple relationship between the two variables. Nevertheless, the relationship of ES and EE performance with GDP remained constant or showed a slightly negative sign. In contrast, ESUS performance tends to increase in higher-income countries. Therefore, rich countries typically perform better on environmental sustainability than poor countries.

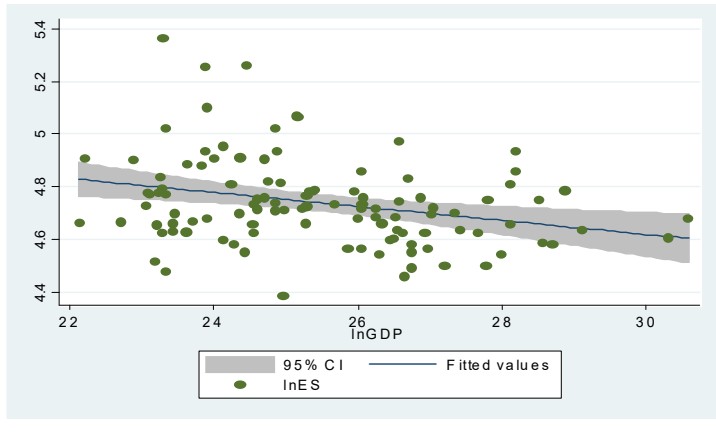

(**a**) Energy security (ES) vs. GDP

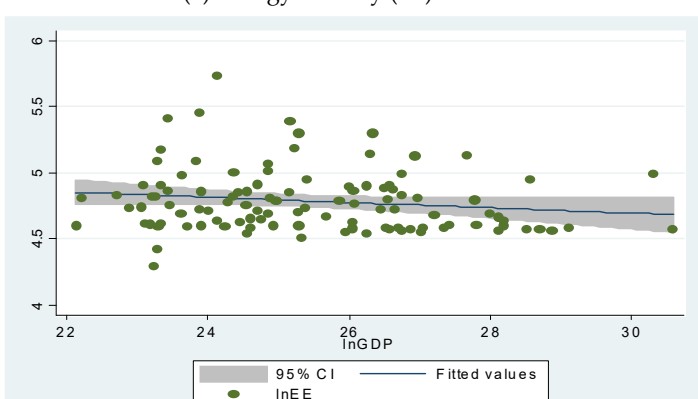

(**b**) Energy equity (EE) vs. GDP

**Figure 3.** *Cont.*

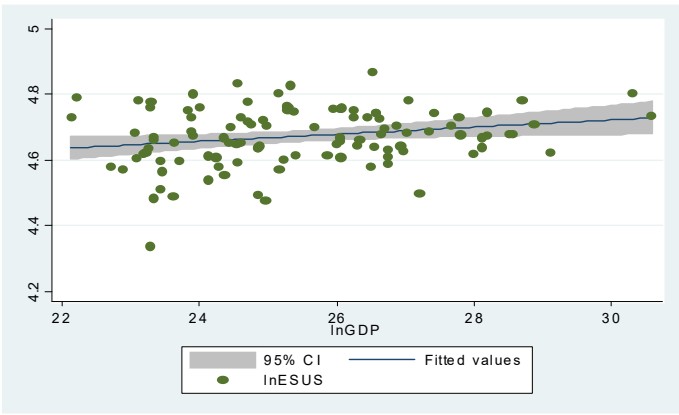

(**c**) Environmental sustainability (ESUS) vs. GDP

**Figure 3.** Three dimensions of energy trilemma and GDP in 2020 (full sample).

### 3.2. Model of Energy Trilemma with Economic Growth

Understanding the energy-growth nexus has crucial implications for policymakers [4]. Therefore, this study examined the relationship between energy and economic growth before constructing economic models. Table 3 displays the general perspective of the energy-growth nexus based on four representative hypotheses. If energy is positively related or positively feedbacks with growth, we denote these by the "growth hypothesis" or "feedback hypothesis". Otherwise, it implies "neutrality" or "conservation hypothesis". Another common hypothesis of the energy-growth nexus is that economic growth shifts households toward better fuel, that is, the "energy ladder" [27]. Thus, this study considered the various hypotheses regarding the energy-growth nexus and constructed the following economic model.

**Table 3.** General perspective of the energy-output nexus.

| Hypothesis | Meaning and Policy Implication |
|---|---|
| Neutrality | -No relationship between energy consumption and output a particular nation<br>-Neither conservative nor established expansive energy policies have any effect on economic output |
| Growth hypothesis | -Unidirectional causality from energy use to economic output<br>-Any regulations on the amount of energy to be consumed will have an effect on the overall growth and development |
| Conservation hypothesis | -Unidirectional causality from economic output to energy consumption<br>-Approach to reduce the energy demand has little or no impact at all on economic output |
| Feedback hypothesis | -Bidirectional causality between energy use and economic growth<br>-Both variables are inseparable as each one of them has simultaneous impact on the other |

Source: Reprinted from [4].

To analyze the relationship between ET and economic growth, this study utilized the following extended Cobb-Douglas production function, adopted from previous studies [4,6,28,29]:

$$Y = f(A, L, K, ET) \tag{1}$$

where Y is the gross domestic product (GDP), A is technology, L is labor, K is capital, and ET is the "energy trilemma". In various studies, technology in Equation (1) can be endogenously determined by the levels of trade openness and financial development [4]. ET can be divided into three aspects- energy security, energy equity, and environmental sustainability- with respect to the ET index. Khan et al. [6] created a new index using PCA, while Le and Nguyen [4] utilized the ET index dimensions individually. Although PCA is convenient for constructing and analyzing of model, it has the disadvantage of not reflecting the various dimensions of ET. Therefore, in this study, ET was separated from

the three aspects and included separately in the model. The detailed model using a log transformation of Equation (1) is as follows:

$$\ln(GDP_{i\,t}) = a_0 + a_1 \ln(L_{i\,t}) + a_2 \ln(K_{i\,t}) + a_3 \ln(T_{i\,t}) + a_4 \ln(ES_{i\,t}) + a_5 \ln(EE_{i\,t}) + a_6 \ln(ESUS_{i\,t}) + b_i + \mu_{i\,t} \tag{2}$$

where T is trade openness, ES is the energy security index, EE is the energy equity index, ESUS is the energy sustainability index, $b_i$ ($i = 1, \ldots, n$) is the unknown intercept for each country and $\mu_{i\,t}$ is the error term.

Based on panel data, this study adopted various types of regression for Equation (2) such as pooled OLS, FGLS, fixed/random effects and GMM. The pooled OLS needs to qualify for homogeneity across panel groups. Therefore, the FGLS estimates the structure of heteroskedasticity from pooled OLS, even if the FGLS does not guarantee unbiased estimation. The fixed effects investigate the relationship between the outcome and predictor variables within a group, and the random effects assume error terms within the model. In addition, this study applied the Hausman and Pesaran tests on these panel analyses to determine whether the unique errors were correlated with regressors or across entities.

Note that the GMM approach allows us to consider other crucial and observable country characteristics and apply it to estimate the dynamic panel data model, although these characteristics are not strictly exogenous [30]. To apply GMM in Equation (2), this study reconstructed it as follows:

$$\ln(GDP_{i\,t}) = c_0 + c_1 \ln(GDP_{i\,t-1}) + c_2 \ln(L_{i\,t}) + c_3 \ln(K_{i\,t}) + c_4 \ln(T_{i\,t}) + c_5 \ln(ES_{i\,t}) + c_6 \ln(EE_{i\,t}) + c_7 \ln(ESUS_{i\,t}) + d_t + d_i + \theta_{i\,t} \tag{3}$$

where $GDP_{i\,t-1}$ is the GDP of the $t-1$ period, $d_t$ is the time-specific effect that considers shocks common to all countries, $d_i$ is an individual-specific effect, and $\theta_{i\,t}$ is the error term. The GMM approach utilizes lagged instruments based on the differenced-GMM (DIF-GMM) suggested by Arellano and Bond [31] and the system-GMM (SYS-GMM) proposed by Blundell and Bond [32]. Likewise, this study tested for serial correlation with respect to first and second-order serial correlation (Arellano-Bond test for AR(1) and AR(2)), including the null hypothesis of no autocorrelation, and adopted the Sargan test to check for over-identifying restrictions.

Based on the feedback hypothesis between energy and economic growth, this study utilized a simultaneous equation model to provide the spatial effects of assessing the nexus [33]. Therefore, this study examined three-way linkages using the following simultaneous equation model among the three variables (ES, EE, and ESUS):

$$\ln(ES_{i\,t}) = \gamma_0 + \gamma_1 \ln(L_{i\,t}) + \gamma_2 \ln(K_{i\,t}) + \gamma_3 \ln(T_{i\,t}) + \gamma_4 \ln(GDP_{i\,t}) + \sigma_{1i\,t} \tag{4}$$

$$\ln(EE_{i\,t}) = \delta_0 + \delta_1 \ln(L_{i\,t}) + \delta_2 \ln(K_{i\,t}) + \delta_3 \ln(T_{i\,t}) + \delta_4 \ln(GDP_{i\,t}) + \sigma_{2i\,t} \tag{5}$$

$$\ln(ESUS_{i\,t}) = \varepsilon_0 + \varepsilon_1 \ln(L_{i\,t}) + \varepsilon_2 \ln(K_{i\,t}) + \varepsilon_3 \ln(T_{i\,t}) + \varepsilon_4 \ln(GDP_{i\,t}) + \sigma_{3i\,t} \tag{6}$$

Hundie and Daksa [34] and Zhang et al. [35] highlighted the energy-environmental Kuznets curve (EKC) hypothesis whereby energy security is influenced by economic growth. These authors found that energy security initially increases with economic growth and then decreases. That is, the energy-EKC hypothesis has a pattern comparable to that of the traditional EKC hypothesis and implies that the degree of energy security can vary depending on the stage of economic growth. Based on the energy-EKC hypothesis, Equation (4) investigates the effects of GDP on ES, where economic growth can contribute to increasing energy security performance. Next, Equation (5) explores the effects of GDP on EE, where economic growth can improve energy equity. Lastly, Equation (6) denotes the effects of GDP on ESUS, where economic growth fosters environmental sustainability.

Finally, this study also adopted time-series analysis that characterized the response variable with respect to time sequenced data from 2000 to 2020. Unlike the panel analysis,

this procedure investigated the relationship between WETI and economic growth through OLS for each country, classifying them by income levels and regions.

*3.3. Descriptive Data*

This study used panel and time series data set on 109 countries from 2000 to 2020. Table 4 lists the estimated variables, definitions, data sources, macroeconomic data (GDP, L, K, and T) collected from the World Bank Open database and WETI (EE, ES, and ESUS) from WEC.

**Table 4.** Definition and source of the estimated variables.

| Variable | Definition (Units) | Source |
|---|---|---|
| GDP | GDP constant 2015 (US $) | World Bank Open Data (https://data.worldbank.org/, (accessed on 20 January 2022)) |
| L | Labor force (total) | |
| K | Gross capital formation constant 2015 (US $) | |
| T | Trade (% of GDP) | |
| ES | Energy security index (2000 = 100) | World Energy Council Energy Trilemma Index (https://trilemma.worldenergy.org/#!/energy-index, (accessed on 20 January 2022)) |
| EE | Energy equity index (2000 = 100) | |
| ESUS | Environmental sustainability index (2000 = 100) | |

Table 5 presents a list of sampled countries. This list comes from the World Bank; in fiscal 2022, it originally included 217 countries (27 low-income countries, 55 lower- and upper-middle income countries each, 80 high income countries). However, due to missing data, this study uses only 109 countries, and income level groups by country are organized into three sub-samples groups (low and lower-middle (hereafter, also "low/lower-middle"), upper-middle-, and high-income countries). Finally, Table 6 illustrates the descriptive statistics of each variable, including "overall", "between", and "within" of the panel data set.

**Table 5.** Country list.

| Income Levels | Country | Number |
|---|---|---|
| Low-income countries ($1045 or less by GNI per capita) | Chad, Congo, Dem. Rep, Ethiopia, Madagascar, Mozambique, Niger | 6 |
| Lower middle-income countries ($1046 to $4095 by GNI per capita) | Angola, Algeria, Bangladesh, Benin, Bolivia, Cambodia, Cameroon, Cote d'Ivoire, Egypt, El Salvador, Eswatini, Ghana, Honduras, India, Indonesia, Iran, Kenya, Mauritania, Mongolia, Morocco, Nepal, Nicaragua, Nigeria, Pakistan, Philippines, Senegal, Sri Lanka, Tanzania, Tunisia, Ukraine, Vietnam | 31 |
| Upper middle-income countries ($4096 to $12,695 by GNI per capita) | Albania, Argentina, Armenia, Bosnia and Herzegovina, Botswana, Brazil, Bulgaria, China, Colombia, Costa Rica, Dominican Rep, Ecuador, Gabon, Guatemala, Jamaica, Jordan, Kazakhstan, Lebanon, Malaysia, Mauritius, Mexico, Moldova, Montenegro, Namibia, North Macedonia, Panama, Paraguay, Peru, Romania, Russia, Serbia, South Africa | 32 |
| High income countries ($12,696 or more by GNI per capita) | Austria, Bahrain, Belgium, Brunei, Canada, Chile, Cyprus, Czech Rep, Denmark, Estonia, Finland, France, Germany, Greece, Hungary, Iceland, Israel, Italy, Japan, Korea Rep, Latvia, Lithuania, Luxembourg, Malta, Netherland, New Zealand, Norway, Oman, Poland, Portugal, Saudi Arabia, Slovak Rep, Slovenia, Spain, Sweden, Switzerland, United Arab Emirates, United Kingdom, United States, Uruguay | 40 |
| Total | | 109 |

**Table 6.** Data descriptive.

| | | | GDP | L | K | T | ES | EE | ESUS |
|---|---|---|---|---|---|---|---|---|---|
| **Full sample** (N = 2289, n = 109, T = 21) | Mean | | $5.56 \times 10^{11}$ | $2.59 \times 10^{7}$ | $1.40 \times 10^{11}$ | 84.17 | 105.12 | 110.40 | 104.12 |
| | Std. Dev. | Overall | $1.89 \times 10^{12}$ | $8.68 \times 10^{7}$ | $5.05 \times 10^{11}$ | 45.53 | 13.02 | 22.75 | 9.10 |
| | | Between | $1.85 \times 10^{12}$ | $8.71 \times 10^{7}$ | $4.72 \times 10^{11}$ | 43.40 | 7.96 | 16.08 | 6.85 |
| | | Within | $4.19 \times 10^{11}$ | 3,951,747 | $1.85 \times 10^{11}$ | 14.35 | 10.33 | 16.16 | 6.02 |
| | Min | Overall | $2.48 \times 10^{9}$ | 156,006 | $4.40 \times 10^{8}$ | 19.55 | 75.04 | 48.54 | 59.63 |
| | | Between | $3.50 \times 10^{9}$ | 188,832.2 | $5.26 \times 10^{8}$ | 26.21 | 88.98 | 80.42 | 79.76 |
| | | Within | $-4.65 \times 10^{12}$ | $-3.39 \times 10^{7}$ | $-2.46 \times 10^{12}$ | 19.85 | 74.22 | 9.10 | 82.66 |
| | Max | Overall | $2.00 \times 10^{13}$ | $7.93 \times 10^{8}$ | $6.37 \times 10^{12}$ | 380.10 | 213.43 | 310.53 | 135.19 |
| | | Between | $1.67 \times 10^{13}$ | $7.73 \times 10^{8}$ | $3.45 \times 10^{12}$ | 390.29 | 130.90 | 201.30 | 121.46 |
| | | Within | $7.22 \times 10^{12}$ | $6.39 \times 10^{7}$ | $3.21 \times 10^{12}$ | 154.98 | 196.84 | 219.63 | 125.49 |
| **Low & lower middle income countries** (N = 777, n = 37, T = 21) | Mean | | $1.37 \times 10^{11}$ | $3.11 \times 10^{7}$ | $4.27 \times 10^{10}$ | 69.43 | 104.95 | 121.54 | 101.97 |
| | Std. Dev. | Overall | $3.04 \times 10^{11}$ | $7.84 \times 10^{7}$ | $1.11 \times 10^{11}$ | 31.90 | 12.83 | 34.05 | 9.50 |
| | | Between | $2.87 \times 10^{11}$ | $7.55 \times 10^{7}$ | $1.01 \times 10^{11}$ | 29.21 | 8.05 | 21.95 | 7.24 |
| | | Within | $1.11 \times 10^{11}$ | 5,705,248 | $4.80 \times 10^{10}$ | 13.64 | 10.06 | 26.26 | 6.25 |
| | Min | Overall | $2.48 \times 10^{9}$ | 290,519 | $4.40 \times 10^{8}$ | 20.72 | 82.25 | 48.54 | 59.63 |
| | | Between | $3.50 \times 10^{9}$ | 328,115.3 | $5.26 \times 10^{8}$ | 30.23 | 92.15 | 80.42 | 79.76 |
| | | Within | $-6.79 \times 10^{11}$ | $-2.88 \times 10^{7}$ | $-3.02 \times 10^{11}$ | 23.31 | 75.16 | 20.23 | 81.84 |
| | Max | Overall | $2.70 \times 10^{12}$ | $4.95 \times 10^{8}$ | $8.71 \times 10^{11}$ | 211.49 | 197.92 | 310.53 | 127.38 |
| | | Between | $1.62 \times 10^{12}$ | $4.57 \times 10^{8}$ | $5.17 \times 10^{11}$ | 157.53 | 129.78 | 201.30 | 117.56 |
| | | Within | $1.22 \times 10^{12}$ | $6.91 \times 10^{7}$ | $5.22 \times 10^{11}$ | 133.87 | 173.79 | 230.76 | 122.21 |
| **Upper-middle income countries** (N = 672, n = 32, T = 21) | Mean | | $4.42 \times 10^{11}$ | $3.56 \times 10^{7}$ | $1.43 \times 10^{11}$ | 78.59 | 104.75 | 109.30 | 104.34 |
| | Std. Dev. | Overall | $1.56 \times 10^{12}$ | $1.34 \times 10^{8}$ | $6.58 \times 10^{11}$ | 32.50 | 12.42 | 11.51 | 9.33 |
| | | Between | $1.43 \times 10^{12}$ | $1.36 \times 10^{8}$ | $5.81 \times 10^{11}$ | 29.66 | 8.38 | 7.41 | 7.04 |
| | | Within | $6.83 \times 10^{11}$ | 3,683,355 | $3.26 \times 10^{11}$ | 14.22 | 9.28 | 8.90 | 6.24 |
| | Min | Overall | $2.68 \times 10^{9}$ | 234,186 | $5.43 \times 10^{8}$ | 21.85 | 75.04 | 88.12 | 74.84 |
| | | Between | $3.64 \times 10^{9}$ | 250,806 | $8.88 \times 10^{8}$ | 26.21 | 91.61 | 97.30 | 86.51 |
| | | Within | $-4.76 \times 10^{12}$ | $-2,403,450$ | $-2.46 \times 10^{12}$ | 20.34 | 73.84 | 82.71 | 82.87 |
| | Max | Overall | $1.46 \times 10^{13}$ | $7.93 \times 10^{8}$ | $6.37 \times 10^{12}$ | 220.40 | 193.23 | 147.73 | 135.19 |
| | | Between | $7.97 \times 10^{12}$ | $7.73 \times 10^{8}$ | $3.30 \times 10^{12}$ | 165.18 | 130.90 | 124.12 | 121.46 |
| | | Within | $7.10 \times 10^{12}$ | $5.56 \times 10^{7}$ | $3.21 \times 10^{12}$ | 133.82 | 167.07 | 132.91 | 125.71 |
| **High income countries** (N = 840, n = 40, T = 21) | Mean | | $1.04 \times 10^{12}$ | $1.33 \times 10^{7}$ | $2.26 \times 10^{11}$ | 102.27 | 105.58 | 100.98 | 105.94 |
| | Std. Dev. | Overall | $2.70 \times 10^{12}$ | $2.68 \times 10^{7}$ | $5.67 \times 10^{11}$ | 57.57 | 13.65 | 5.31 | 8.09 |
| | | Between | $2.7 \times 10^{12}$ | $2.71 \times 10^{7}$ | $5.68 \times 10^{11}$ | 56.23 | 7.71 | 4.24 | 5.87 |
| | | Within | $3.06 \times 10^{11}$ | 1,276,862 | $7.83 \times 10^{10}$ | 15.09 | 11.32 | 3.27 | 5.63 |
| | Min | Overall | $6.63 \times 10^{9}$ | 156,006 | $1.01 \times 10^{9}$ | 19.55 | 76.33 | 88.99 | 75.46 |
| | | Between | $9.23 \times 10^{9}$ | 188,832.2 | $1.92 \times 10^{9}$ | 26.75 | 89.98 | 96.73 | 88.31 |
| | | Within | $-1.94 \times 10^{12}$ | 3,199,535 | $-5.09 \times 10^{11}$ | 37.95 | 75.75 | 78.71 | 89.16 |
| | Max | Overall | $2.00 \times 10^{13}$ | $1.67 \times 10^{8}$ | $4.33 \times 10^{12}$ | 380.10 | 213.43 | 137.99 | 132.02 |
| | | Between | $1.67 \times 10^{13}$ | $1.57 \times 10^{8}$ | $3.45 \times 10^{12}$ | 309.29 | 121.71 | 122.27 | 116.78 |
| | | Within | $4.30 \times 10^{12}$ | $2.38 \times 10^{7}$ | $1.11 \times 10^{12}$ | 173.08 | 197.29 | 116.70 | 125.35 |

## 4. Empirical Results

### 4.1. Panel Analysis Results

Tables 7–10 indicate the estimation results of the panel analysis with respect to the pooled OLS, FGLS, fixed effects, DIF-GMM, and SYS-GMM, respectively. Before explaining the results of panel analysis, this study adopted several kinds of panel model tests including the Hausman, Pasaran, Arellano-Bond, and Sargan tests.

**Table 7.** Estimation results (full sample).

|  | POLS | FGLS | Fixed Effect | DIF-GMM | SYS-GMM |
|---|---|---|---|---|---|
| $\ln(GDP)_{t-1}$ |  |  |  | 0.77 *** (9.68) | 0.87 *** (8.42) |
| $\ln(L)$ | 0.01 (1.06) | 0.01 *** (3.65) | 0.43 *** (19.40) | 0.09 *** (6.19) | 0.01 *** (3.70) |
| $\ln(K)$ | 0.94 *** (184.55) | 0.92 *** (205.41) | 0.35 *** (40.87) | 0.10 *** (23.34) | 0.09 *** (21.92) |
| $\ln(T)$ | −0.17 *** (−10.44) | −0.15 *** (−13.36) | −0.03 ** (−2.51) | 0.06 *** (12.11) | 0.05 *** (9.62) |
| $\ln(ES)$ | 0.02 (0.42) | 0.01 (0.63) | 0.32 *** (11.72) | 0.04 *** (3.88) | 0.02 *** (2.62) |
| $\ln(EE)$ | 0.42 *** (11.55) | 0.41 *** (27.65) | 0.40 *** (18.58) | 0.03 *** (3.51) | 0.05 *** (7.52) |
| $\ln(ESUS)$ | 0.92 *** (13.01) | 0.81 *** (17.51) | 0.26 *** (6.79) | −0.03 *** (−3.71) | −0.08 *** (−6.16) |
| constant | 1.27 *** (3.19) | 1.71 *** (7.38) | 5.57 *** (6.79) | 1.37 *** (12.48) | 1.35 *** (16.26) |
| $R^2$ | 0.98 |  | 0.82 |  |  |
| Observation | 2289 | 2289 | 2289 | 2071 | 2180 |
| Hausman |  | Chi2(6) = 50.24, Prob > Chi2 = 0.00 |  |  |  |
| Pasaran |  | Pr = 0.37 |  |  |  |
| AR(1) |  |  |  | −5.15 *** Prob > z = 0.00 | −5.39 *** Prob > z = 0.00 |
| AR(2) |  |  |  | −0.51 Prob > z = 0.60 | −0.35 Prob > z = 0.72 |
| Sargan |  |  |  | Prob > Chi2 = 0.21 | Prob > Chi2 = 0.19 |

Note: ** and *** indicate the significant level of 10%, 5%, and 1%, respectively. The numbers in parentheses are t−Values.

**Table 8.** Estimation results (low/lower-middle income countries).

|  | POLS | FGLS | Fixed Effect | DIF-GMM | SYS-GMM |
|---|---|---|---|---|---|
| $\ln(GDP)_{t-1}$ |  |  |  | 0.82 *** (52.05) | 0.89 *** (11.87) |
| $\ln(L)$ | 0.14 *** (8.59) | 0.15 *** (36.11) | 1.14 *** (21.89) | 0.15 *** (4.56) | 0.03 *** (6.13) |
| $\ln(K)$ | 0.77 *** (56.31) | 0.76 *** (233.32) | 0.21 *** (14.31) | 0.07 *** (12.54) | 0.07 *** (13.53) |
| $\ln(T)$ | −0.22 *** (−6.77) | −0.21 *** (−39.62) | −0.06 *** (−2.66) | 0.02 ** (2.55) | 0.02 *** (3.30) |

**Table 8.** *Cont.*

|  | POLS | FGLS | Fixed Effect | DIF-GMM | SYS-GMM |
|---|---|---|---|---|---|
| ln(ES) | 0.08 (0.69) | 0.06 ** (2.22) | 0.15 *** (2.69) | 0.05 *** (2.82) | 0.08 *** (4.42) |
| ln(EE) | −0.12 ** (−2.55) | −0.12 *** (−11.77) | −0.23 *** (−8.38) | −0.81 (−0.81) | −0.02 *** (−2.98) |
| ln(ESUS) | 1.20 *** (9.44) | 1.18 *** (49.77) | −0.01 (−0.15) | −0.03 (−1.51) | −0.05 *** (−2.90) |
| constant | 0.67 (0.85) | 0.66 *** (5.38) | −0.57 (−0.85) | 0.41 * (1.81) | 0.99 *** (7.60) |
| $R^2$ | 0.94 | | 0.86 | | |
| Observation | 777 | 777 | 777 | 703 | 740 |
| Hausman | | | Chi2(6) = 41.11 Prob > Chi2 = 0.00 | | |
| Pasaran | | | Pr = 0.34 | | |
| AR(1) | | | | −3.08 *** Prob > z = 0.00 | −3.47 *** Prob > z = 0.00 |
| AR(2) | | | | −1.27 Prob > z = 0.20 | −1.24 Prob > z = 0.21 |
| Sargan | | | | Prob > Chi2 = 0.30 | Prob > Chi2 = 0.34 |

Note: *, ** and *** indicate the significant level of 10%, 5%, and 1%, respectively. The numbers in parentheses are t−Values.

**Table 9.** Estimation results (upper-middle income countries).

|  | POLS | FGLS | Fixed Effect | DIF-GMM | SYS-GMM |
|---|---|---|---|---|---|
| $\ln(GDP)_{t-1}$ | | | | 0.79 *** (8.60) | 0.85 *** (8.14) |
| ln(L) | 0.18 *** (8.91) | 0.19 *** (58.20) | 0.21 *** (5.77) | 0.02 (1.17) | 0.07 *** (8.89) |
| ln(K) | 0.79 *** (43.05) | 0.78 *** (280.62) | 0.41 *** (31.58) | 0.10 *** (13.99) | 0.07 *** (10.71) |
| ln(T) | −0.21 *** (−8.51) | −0.20 *** (−46.83) | −0.08 *** (−3.97) | 0.08 *** (7.49) | 0.10 *** (9.70) |
| ln(ES) | 0.22 *** (2.82) | 0.20 *** (16.89) | 0.18 *** (3.89) | 0.06 *** (3.14) | 0.04 *** (3.22) |
| ln(EE) | −0.57 *** (−6.56) | −0.55 *** (−42.34) | −0.90 *** (−14.60) | 0.09 (1.24) | −0.07 *** (−3.24) |
| ln(ESUS) | 0.29 *** (3.16) | 0.30 *** (29.66) | 0.38 *** (6.80) | 0.01 (0.44) | −0.04 *** (−2.23) |
| constant | 4.65 *** (7.29) | 4.68 *** (5.95) | 5.30 *** (10.33) | 1.36 *** (5.91) | 0.68 *** (4.45) |
| $R^2$ | 0.98 | | 0.88 | | |
| Observation | 672 | 672 | 672 | 608 | 640 |
| Hausman | | | Chi2(6) = 89.59 Prob > Chi2 = 0.00 | | |
| Pasaran | | | Pr = 0.51 | | |
| AR(1) | | | | −3.07 *** Prob > z = 0.00 | −2.90 *** Prob > z = 0.00 |

**Table 9.** *Cont.*

| | POLS | FGLS | Fixed Effect | DIF-GMM | SYS-GMM |
|---|---|---|---|---|---|
| AR(2) | | | | −1.47<br>Prob > z = 0.14 | −1.38<br>Prob > z = 0.16 |
| Sargan | | | | Prob > Chi2 = 0.48 | Prob > Chi2 = 0.36 |

Note: *** indicate the significant level of 10%, 5%, and 1%, respectively. The numbers in parentheses are t−Values.

**Table 10.** Estimation results (high income countries).

| | POLS | FGLS | Fixed Effect | DIF-GMM | SYS-GMM |
|---|---|---|---|---|---|
| $\ln(\text{GDP})_{t-1}$ | | | | 0.67 ***<br>(5.78) | 0.77 ***<br>(8.29) |
| $\ln(L)$ | 0.11 ***<br>(8.02) | 0.12 ***<br>(32.75) | 0.21 ***<br>(9.30) | 0.05 ***<br>(3.80) | 0.11 ***<br>(10.10) |
| $\ln(K)$ | 0.87 ***<br>(71.83) | 0.84 ***<br>(49.44) | 0.35 ***<br>(26.54) | 0.13 ***<br>(23.19) | 0.12 ***<br>(21.96) |
| $\ln(T)$ | 0.04 **<br>(2.40) | 0.04 ***<br>(9.41) | 0.21 ***<br>(10.28) | 0.01 ***<br>(13.32) | 0.01 **<br>(2.41) |
| $\ln(ES)$ | 0.06<br>(1.03) | 0.07 ***<br>(4.53) | 0.33 ***<br>(10.94) | 0.10 ***<br>(8.66) | 0.02 **<br>(2.06) |
| $\ln(EE)$ | 1.05 ***<br>(7.21) | 1.05 ***<br>(5.39) | 0.39 ***<br>(4.64) | 0.12 ***<br>(3.18) | 0.21 ***<br>(5.86) |
| $\ln(ESUS)$ | 0.40 ***<br>(3.90) | 0.39 ***<br>(19.27) | 0.22 ***<br>(4.13) | 0.05 ***<br>(2.76) | 0.11 ***<br>(5.78) |
| constant | 6.34 ***<br>(7.34) | 6.37 ***<br>(5.67) | 8.83 ***<br>(18.90) | 3.10 ***<br>(16.63) | 2.27 ***<br>(11.98) |
| $R^2$ | 0.98 | | 0.80 | | |
| Observation | 840 | 840 | 840 | 760 | 800 |
| Hausman | | | Chi2(6) = 72.61<br>Prob > Chi2 = 0.00 | | |
| Pasaran | | | Pr = 0.22 | | |
| AR(1) | | | | −4.59 ***<br>Prob > z = 0.00 | −3.86 ***<br>Prob > z = 0.00 |
| AR(2) | | | | −0.54<br>Prob > z = 0.58 | −1.40<br>Prob > z = 0.16 |
| Sargan | | | | Prob > Chi2 = 0.29 | Prob > Chi2 = 0.13 |

Note: ** and *** indicate the significant level of 10%, 5%, and 1%, respectively. The numbers in parentheses are t−Values.

First, the Hausman test determines an appropriate estimator between fixed and random effects, including the null hypothesis of the non-systematic difference in coefficients. All Hausman test results from Tables 7–10 indicate that the null hypothesis can be rejected. That is, fixed effects are preferred to random effects. Fixed effects are a useful estimator if we are only interested in analyzing the impact of variables that vary over time. In addition, fixed effects remove influence of time-invariant characteristics so that we can assess the net effect of the predictors on the dependent variable [36].

Second, the Pasaran test provides meaningful results if cross-sectional dependence is a problem in macro panels with long time series, with the null hypothesis that the residuals across entities are not correlated. All Pasaran test results from Tables 7–10 show that the null hypothesis cannot be rejected, implying that there is no cross-sectional dependence.

Third, the pre-tests of dynamic panel analysis for GMM were applied using Arellano-Bond and Sargan tests. The Arellano-Bond test includes AR(1) and AR(2) which serial

correlation has first and second-order, respectively. In Tables 7–10, all *p*-values of the AR tests indicate the presence of serial correlation in the first order but not in the second-order. That is AR(1) is rejected, but AR(2) is not rejected. In addition, the Sargan test includes the null hypothesis that over-identifying restrictions are valid. Likewise, all results of the Sargan test do not reject the null hypothesis, implying that we do not need to reconsider the model.

Table 7 presents the estimation results of the panel series analysis for the full sample. L and K have positive effects on GDP, implying that labor and capital, as major production factors in the traditional production function, contribute to economic growth in the full sample. However, T indicated different results depending on the panel analysis method. For example, trade has a positive effect on economic growth in the case of dynamic panel analysis for GMM. Better ES and EE performances contribute to economic growth. However, the effect of ESUS performance on economic growth differs, depending on the panel analysis methods. In a dynamic relationship, an increase in ESUS performance is negatively related to economic growth. Therefore, the panel analysis model shows that ET and economic growth have a complementary relationship in the full sample.

Tables 8 and 9 present the estimation results of the panel analysis for low/lower-middle and upper-middle income countries. Similar to the full sample, the coefficients of L, K, and T positively affect economic growth. ES has a positive effect on economic growth, while EE and ESUS performance have negative effects. Therefore, in the cases of low/lower-middle and upper-middle income countries, increased of EE and ESUS performance both negatively effects economic growth, while ES performance contributes to economic growth. These results imply a trade-off between ET and economic growth.

Table 10 illustrates the estimation results of the panel series analysis for high income countries. L, K, T, and GDP from the t−1 period positively contribute to economic growth. Unlike the previous two sub-sample cases (low/lower-middle and upper-middle income countries), all dimensions of ET have positive effects on economic growth. Thus, high income countries do not exhibit a trade-off between ET and economic growth. This implies that these countries have relatively consistently implemented current energy-related policies over a long period of time. Moreover, this can be evidence that high income countries have started to improve energy efficiency in society as a whole and converted to a low-carbon industry faster than countries with other income-levels.

Table 11 presents the estimation results for the simultaneous equation models. In the full sample, as economic growth progressed, ESUS performance increased, but EE decreased. In particular, economic growth in low/lower-middle income countries positively affected ESUS performance, while it had no significant effect on ES and EE. However, in upper-middle- and high-income countries, GDP positively affected ES and ESUS, but negatively affected EE. Therefore, we found that ES, EE, and ESUS performance have different aspects depending on the income levels of each country.

**Table 11.** Estimation results (simultaneous equations).

| | | Dependent Variable | | |
| --- | --- | --- | --- | --- |
| | | **ln(ES)** | **ln(EE)** | **ln(ESUS)** |
| | ln(GDP) | 0.01 (1.02) | −0.12 *** (−10.85) | 0.07 *** (12.52) |
| | ln(L) | 0.01 (0.53) | 0.03 *** (9.99) | −0.01 (−0.20) |
| | ln(K) | −0.01 (−1.41) | 0.08 *** (7.74) | −0.05 *** (−10.43) |
| Full sample | ln(T) | 0.01 *** (3.17) | −0.01 * (−1.93) | 0.03 *** (7.77) |
| | constant | 4.59 *** (85.15) | 5.17 *** (64.52) | 4.08 *** (97.95) |
| | $R^2$ | 0.01 | 0.11 | 0.08 |

**Table 11.** *Cont.*

| | | Dependent Variable | | |
|---|---|---|---|---|
| | | **ln(ES)** | **ln(EE)** | **ln(ESUS)** |
| Low and lower-middle income countries | ln(GDP) | −0.01 (−1.15) | −0.02 (−1.17) | 0.08 *** (9.10) |
| | ln(L) | 0.01 (0.53) | −0.03 ** (−2.59) | −0.01 (−1.57) |
| | ln(K) | 0.01 (0.76) | 0.06 *** (3.12) | −0.06 *** (−7.95) |
| | ln(T) | 0.01 * (1.82) | −0.02 (−1.13) | 0.05 *** (5.69) |
| | constant | 4.66 *** (48.10) | 4.58 *** (20.1) | 3.93 *** (47.55) |
| | $R^2$ | 0.01 | 0.02 | 0.11 |
| Upper-middle income countries | ln(GDP) | 0.05 *** (2.72) | −0.10 *** (−6.50) | 0.06 *** (4.52) |
| | ln(L) | −0.03 *** (−3.62) | −0.04 *** (−4.74) | −0.01 (−0.58) |
| | ln(K) | −0.01 (−1.12) | 0.14 *** (10.31) | −0.05 *** (−4.13) |
| | ln(T) | 0.01 (0.39) | −0.04 (−3.54) | 0.02 ** (2.54) |
| | constant | 4.39 *** (33.17) | 4.67 *** (40.35) | 4.19 *** (37.91) |
| | $R^2$ | 0.03 | 0.16 | 0.03 |
| High income countries | ln(GDP) | 0.02 ** (2.33) | −0.05 *** (−7.56) | 0.05 *** (4.47) |
| | ln(L) | 0.03 *** (4.23) | 0.02 *** (8.54) | 0.02 *** (4.58) |
| | ln(K) | −0.05 *** (−3.06) | 0.03 *** (4.38) | −0.06 *** (−5.19) |
| | ln(T) | 0.02 ** (2.49) | 0.01 (1.02) | 0.38 *** (5.76) |
| | constant | 4.74 *** (40.96) | 4.89 *** (106.25) | 4.20 *** (55.67) |
| | $R^2$ | 0.03 | 0.11 | 0.09 |

Note: *, **, and *** indicate the significant level of 10%, 5%, and 1%, respectively. The numbers in parentheses are t−values.

*4.2. Time Series Analysis Results*

Table 12 (Table 12 is summarized by Appendix A, which includes estimation results of the time series analysis for each country by income level) shows the simple relationship between the ET's effects on economic growth for the time series analysis by each country. In 37.83% low and lower-middle income countries (14 countries: Angola, Cote d'Ivoire, Egypt, Eswatini, Ethiopia, Indonesia, Iran, Mongolia, Morocco, Mozambique, Sri Lanka, Tanzania, Ukraine, and Vietnam), the expansion of EE performance contributed to economic growth. Increasing EE and ES performances positively affected economic growth in 31.25% (10 countries: Albania, Armenia, China, Kazakhstan, Lebanon, Mexico, Peru, Romania, Russia, and South Africa) and 28.12% (9 countries: Albania, Brazil, China, Guatemala, Jordan, Malaysia, Montenegro, Romania, and South Africa) upper-middle income countries, respectively. Finally, EE and ESUS positively affected economic growth in 20% (9 countries: Bahrain, Denmark, Germany, Iceland, Japan, Portugal, Slovenia, and Uruguay) and 17.5% (7 countries: Canada, Czech Rep, Denmark, Greece, Hungary, Spain, and Uruguay) high-income countries, respectively.

Finally, this study investigated the relationship between ET and economic growth using time series analysis by region (see Appendix B for more details). The highest proportion of countries where EE performance contributed to economic growth were from Africa and Asia at 8.26% (9 countries: Angola, Egypt, Cote d'Ivoire, Ethiopia, Eswatini, Morocco, Mozambique, Tanzania, and South Africa) and 11.01% (12 countries: Armenia, Bahrain, China, Cyprus, Kazakhstan, Indonesia, Iran, Lebanon, Mongolia, Sri Lanka, United Arab Emirates, and Vietnam), respectively. Meanwhile, the highest proportion of countries where ESUS performance influenced economic growth were from America at 5.5% (6 countries: Canada, Dominican Rep, El Salvador, Guatemala, Peru, and Uruguay). Finally, the highest proportion of countries where ES performance influenced economic growth were from

Europe at 7.34% (8 countries: Albania, Denmark, Germany, Iceland, Montenegro, Romania, Portugal, and Slovenia).

**Table 12.** Simple relationship regarding the effects of energy trilemma on economic growth (time series analysis for each country).

| | ES on Economic Growth | | EE on Economic Growth | | ESUS on Economic Growth | |
|---|---|---|---|---|---|---|
| | **(+)** | **(−)** | **(+)** | **(−)** | **(+)** | **(−)** |
| Low and lower-middle income countries (number of country) | Bangladesh Cambodia Kenya Morocco Nicaragua Niger (6) | Honduras Mozambique (2) | Angola Cote d'Ivoire Egypt Eswatini Ethiopia Indonesia Iran Mongolia Morocco Mozambique Sri Lanka Tanzania Ukraine Vietnam (14) | Cambodia (1) | El Salvador Eswatini Kenya Nigeria (4) | Angola Chad Indonesia Madagascar Nepal Nicaragua Niger (7) |
| % of sub-sample country | 16.21% | 5.41% | 37.83% | 2.70% | 10.81% | 18.91 |
| Upper-middle income countries (number of country) | Albania Brazil China Guatemala Jordan Malaysia Montenegro Romania South Africa (9) | Bosnia and Herzegovina Costa Rica (2) | Albania Armenia China Kazakhstan Lebanon Mexico Peru Romania Russia South Africa (10) | | Dominican Rep Guatemala Jordan Peru (4) | Argentina Bosnia and Herzegovina Ecuador Gabon (4) |
| % of sub-sample country | 28.12% | 6.25% | 31.25% | 0% | 12.5% | 12.5% |
| High income countries (number of country) | Bahrain Denmark Germany Iceland Japan Portugal Slovenia Uruguay (8) | Austria Norway United Arab Emirates (3) | Bahrain Cyprus Estonia Hungary Iceland United Arab Emirates (6) | Belgium (1) | Canada Czech Rep Denmark Greece Hungary Spain Uruguay (7) | Japan Luxembourg United Arab Emirates (3) |
| % of sub-sample country | 20% | 7.5% | 15% | 2.5% | 17.5% | 7.5% |
| Sum (% of full sample country) | 23 (21.1%) | 7 (6.42%) | 30 (27.52%) | 2 (1.83%) | 15 (13.76%) | 14 (12.84%) |

Note: (+) and (−) are selected by estimation results for three sample groups, if they are at least 10% significance level.

In summary, EE performance was conductive to growth in low and lower-middle countries of Asia and Africa and ES performance in high-income countries of. Therefore, the effects of the three dimensions of ET on economic growth vary according to income levels or region.

## 5. Conclusions and Implications

The ET emphasizes that in their energy policy and socio-corporate strategy, all countries need to improve the three dimensions of ET: energy security, energy equity, and

environmental sustainability. These dimensions are all closely related to economic activities, and vice versa. This study analyzed the relationship between ET and economic growth for income levels and regions using extended Cobb-Douglas production function. This study used both panel and time series approaches on data from 109 countries from 2000 to 2020. Regarding the panel regression analysis, this study applied pooled OLS, FGLS, fixed effects, and the dynamic panel approach of GMM, and conducted time series analysis for each country.

The main findings of this study are as follows. For the full sample, both energy security and energy equity had positive relationships with economic growth, while environmental sustainability had a negative relationship with economic growth. Next, we found evidence of an "energy trilemma" in certain income categories of 109 countries. In particular, for high income countries, all three dimensions of ET had positive effects on economic growth. However, for low/lower-middle and upper-middle income countries, energy security positively influenced economic growth. Thus, this latter group of countries face the ET.

Also, in the time series analysis by country and region, this study found that energy equity had the largest impact on economic growth for low/lower-middle income countries, while in high income countries, energy security and energy equity had a positive influence on economic growth. In Africa and Asia, energy equity is helpful in economic growth, but in America and Europe, energy security has contributed to economic growth.

Based on the results of this study, we have two types of suggestions and policy implications. First, the empirical analysis results show the necessity of investigating different income levels, and a country's characteristics and regions, using panel or time-series approaches. That is, especially important considering ET's polygonal aspect, since the factors that affect economic growth differ by country or region.

Second, each country has various kinds of energy demand and supply strategies; therefore, uniform energy-related policies should be avoided. Furthermore, all ET dimensions are necessary for us to live; we cannot achieve sustainable development by neglecting any one of energy security, energy equity, and environmental sustainability. For example, carbon neutrality polices mainly consider environmental sustainability. However, such strategies can be threatened by energy security, demand, and supply, even when environmental protection and climate change are important. Especially, Khan et al. [37] mentioned that increase of balanced ET by 1% contributes to economic growth by 0.38%, and also reduces the ecological footprint by 0.59%. Therefore, we need a stable and comprehensive resource utilization policy, and a balanced ET management policy which considers energy security, energy equity, and environmental sustainability.

**Funding:** This paper was supported by Wonkwang University in 2022.

**Institutional Review Board Statement:** Not applicable.

**Informed Consent Statement:** Not applicable.

**Conflicts of Interest:** The authors declare no conflict of interest.

## Appendix A. Estimation Results of Simple Relationship (Time Series Analysis)

**Table A1.** Low & lower-middle income countries.

| Country | ln(ES) | | ln(EE) | | ln(ESUS) | | $R^2$ |
|---|---|---|---|---|---|---|---|
| | Coefficient | t−Value | Coefficient | t−Value | Coefficient | t−Value | |
| Algeria | 0.64 | 1.66 | 0.22 | 1.3 | −0.35 | −1.2 | 0.98 |
| Angola | −0.26 | −0.92 | 5.49 ** | 2.21 | −0.81 * | −2.1 | 0.97 |
| Bangladesh | 0.16 * | 2.12 | 0.02 | 0.79 | 0.02 | 0.46 | 0.99 |
| Benin | 0.21 | 1.74 | −0.03 | −1.67 | −0.01 | −0.47 | 0.99 |
| Bolivia | 0.71 | 1.06 | 0.33 | 1.2 | −0.22 | −0.79 | 0.98 |

**Table A1.** *Cont.*

| Country | ln(ES) | | ln(EE) | | ln(ESUS) | | R² |
|---|---|---|---|---|---|---|---|
| | Coefficient | t−Value | Coefficient | t−Value | Coefficient | t−Value | |
| Cambodia | 0.10 * | 1.88 | −0.20 ** | −2.24 | −0.03 | −0.29 | 0.99 |
| Cameroon | 0.07 | 0.97 | 0.03 | 0.6 | 0.03 | 0.73 | 0.99 |
| Chad | 1.45 | 1.5 | 0.28 | 1.27 | −0.93 *** | −3.13 | 0.94 |
| Congo | −0.15 | −0.26 | −0.09 | −0.56 | −0.44 | −0.95 | 0.98 |
| Cote d'Ivoire | 0.38 | 0.91 | 0.41 ** | 2.54 | −0.5 | −1.11 | 0.98 |
| Egypt | 0.21 | 0.36 | 0.58 *** | 3.27 | 0.22 | 0.94 | 0.98 |
| El Salvador | −0.06 | −0.42 | 0.15 | 0.97 | 0.67 *** | 3.21 | 0.96 |
| Eswatini | −0.05 | −0.58 | 1.07 ** | 2.96 | 0.16 ** | 2.36 | 0.99 |
| Ethiopia | −0.11 | −0.33 | 0.12 ** | 2.15 | −0.02 | −0.07 | 0.99 |
| Ghana | 0.1 | 0.71 | 0.16 | 0.76 | −0.15 | −0.87 | 0.99 |
| Honduras | −0.46 * | −2.12 | 0.1 | 0.58 | 0.13 | 0.59 | 0.97 |
| India | −0.12 | −0.09 | 0.2 | 1.46 | −0.34 | −0.95 | 0.99 |
| Indonesia | −0.62 | −1.34 | 0.33 ** | 2.25 | −0.57 * | −1.97 | 0.99 |
| Iran | −0.2 | −0.08 | 0.47 * | 2.14 | −0.04 | −0.14 | 0.95 |
| Kenya | 0.28 ** | 2.8 | −0.05 | −1.45 | 0.23 ** | 2.5 | 0.99 |
| Madagascar | 0.03 | 0.11 | 0.08 | 1.35 | −0.29 * | −1.78 | 0.98 |
| Mauritania | 1.43 | 0.67 | −0.1 | −0.87 | −0.07 | −0.18 | 0.97 |
| Mongolia | −1.21 | −0.87 | 0.56 ** | 2.33 | −0.51 | −1.28 | 0.98 |
| Morocco | 0.28 ** | 2.31 | 0.37 *** | 4.41 | −0.04 | −0.1 | 0.99 |
| Mozambique | −0.99 *** | −3.89 | 0.21 ** | 2.19 | −0.03 | −0.13 | 0.98 |
| Nepal | −0.11 | −1.01 | 0.05 | 1.06 | −0.24 ** | −2.79 | 0.99 |
| Nicaragua | 0.13 * | 1.98 | −0.02 | −0.5 | −0.33 ** | −2.83 | 0.99 |
| Niger | 0.28 *** | 4.98 | 0.02 | 1.16 | −0.20 *** | −6.56 | 0.99 |
| Nigeria | 0.02 | 0.02 | −0.01 | −0.13 | 1.25 *** | 3.2 | 0.96 |
| Pakistan | −0.06 | −0.98 | 0.02 | 0.62 | −0.06 | −0.63 | 0.99 |
| Philippines | −0.31 | −0.46 | −0.05 | −0.24 | −0.43 | −1.46 | 0.99 |
| Senegal | −0.03 | −0.71 | 0.04 | 1.25 | −0.07 | −0.85 | 0.99 |
| Sri Lanka | −0.33 | −1.57 | 0.38 *** | 5.14 | −0.01 | −0.05 | 0.99 |
| Tanzania | −0.08 | −1.65 | 0.06 * | 1.85 | −0.11 | −1.59 | 0.99 |
| Tunisia | −0.36 | −1.04 | 0.16 | 0.58 | 0.17 | 0.64 | 0.97 |
| Ukraine | −0.25 | −0.63 | 1.53 *** | 4.44 | 0.51 | 1.32 | 0.86 |
| Vietnam | −0.05 | −0.32 | 0.51 *** | 6.76 | 0.03 | 0.43 | 0.99 |

Note: *, **, and *** indicate the significant level of 10%, 5%, and 1%, respectively. The numbers in parentheses are t−Values.

**Table A2.** Upper-middle income countries.

| Country | ln(ES) | | ln(EE) | | ln(ESUS) | | R² |
|---|---|---|---|---|---|---|---|
| | Coefficient | t−Value | Coefficient | t−Value | Coefficient | t−Value | |
| Albania | 0.51 *** | 3.19 | 1.44 *** | 3.60 | 0.32 | 1.43 | 0.97 |
| Argentina | −0.03 | −0.06 | 0.20 | 1.00 | −0.91 ** | −2.79 | 0.98 |

**Table A2.** *Cont.*

| Country | ln(ES) | | ln(EE) | | ln(ESUS) | | R² |
|---|---|---|---|---|---|---|---|
| | Coefficient | t−Value | Coefficient | t−Value | Coefficient | t−Value | |
| Armenia | −1.39 | −1.70 | 2.31 *** | 3.68 | 0.49 | 0.63 | 0.91 |
| Bosnia and Herzegovina | −0.90 * | −2.11 | 0.60 | 1.29 | −0.30 * | −1.98 | 0.92 |
| Botswana | 0.46 | 0.75 | 0.78 | 0.99 | 0.07 | 0.41 | 0.98 |
| Brazil | 0.24 * | 2.13 | 0.04 | 0.65 | −0.12 | −1.42 | 0.99 |
| Bulgaria | 0.04 | 0.15 | 0.87 | 1.43 | 0.17 | 0.28 | 0.91 |
| China | 1.32 ** | 2.22 | 0.78 *** | 3.27 | 0.12 | 1.30 | 0.99 |
| Colombia | 0.09 | 0.32 | −0.14 | −0.62 | 0.13 | 0.50 | 0.98 |
| Costa Rica | −0.53 * | −1.84 | 0.06 | 0.30 | −0.19 | −0.59 | 0.99 |
| Dominican Rep | 0.01 | 0.03 | 0.11 | 1.01 | 0.30 *** | 3.87 | 0.99 |
| Ecuador | 0.10 | 0.21 | 0.10 | 0.31 | −0.41 * | −1.92 | 0.98 |
| Gabon | 0.29 | 1.55 | 0.54 | 1.69 | −0.27 * | −1.91 | 0.98 |
| Guatemala | 0.41 * | 1.94 | 0.28 | 1.67 | 0.30 ** | 2.65 | 0.99 |
| Jamaica | −0.11 | −1.39 | 0.02 | 0.14 | −0.02 | −0.15 | 0.68 |
| Jordan | 0.25 *** | 3.32 | 0.33 | 1.70 | 0.58 ** | 2.44 | 0.99 |
| Kazakhstan | −0.01 | −0.01 | 0.48 * | 2.05 | 0.20 | 1.21 | 0.99 |
| Lebanon | −0.12 | −1.42 | 0.70 *** | 7.05 | 0.12 | 1.45 | 0.99 |
| Malaysia | 0.67 *** | 3.92 | 0.26 | 1.49 | 0.01 | 0.04 | 0.99 |
| Mauritius | −0.01 | −0.09 | −0.09 | −0.38 | 0.06 | 0.49 | 0.99 |
| Mexico | −0.06 | −0.20 | 0.17 * | 2.04 | 0.05 | 0.22 | 0.98 |
| Moldova | −0.06 | −0.27 | 0.21 | 0.96 | 0.61 | 1.63 | 0.96 |
| Montenegro | 0.61 ** | 2.76 | −0.22 | −0.34 | 0.03 | 0.12 | 0.92 |
| Namibia | −0.02 | −0.07 | 0.04 | 0.03 | −0.03 | −0.10 | 0.98 |
| North Macedonia | 0.05 | 0.24 | −0.01 | −0.03 | 0.03 | 0.16 | 0.98 |
| Panama | −0.02 | −0.10 | 0.20 | 0.81 | −0.28 | −1.25 | 0.99 |
| Paraguay | 0.40 | 0.82 | 0.37 | 1.31 | −0.05 | −0.34 | 0.98 |
| Peru | 0.75 | 1.65 | 0.87 *** | 4.36 | 0.46 ** | 2.17 | 0.99 |
| Romania | 0.91 *** | 3.60 | 0.61 ** | 2.66 | −0.32 | −1.09 | 0.97 |
| Russia | −0.74 | −0.88 | 1.06 *** | 4.98 | 0.09 | 0.21 | 0.98 |
| Serbia | −0.40 | −0.42 | −0.77 | −1.07 | −0.12 | −0.41 | 0.90 |
| South Africa | 0.29 ** | 2.69 | 0.45 *** | 3.37 | 0.12 | 1.01 | 0.98 |

Note: *, **, and *** indicate the significant level of 10%, 5%, and 1%, respectively. The numbers in parentheses are t−Values.

**Table A3.** High income countries.

| Country | ln(ES) | | ln(EE) | | ln(ESUS) | | R² |
|---|---|---|---|---|---|---|---|
| | Coefficient | t−Value | Coefficient | t−Value | Coefficient | t−Value | |
| Austria | −0.21 ** | −2.27 | 0.18 | 0.22 | 0.07 | 0.56 | 0.98 |
| Bahrain | 0.70 *** | 5.28 | 4.48 ** | 2.59 | 0.02 | 0.18 | 0.99 |
| Belgium | 0.11 | 1.46 | −0.95 ** | −2.42 | 0.05 | 0.48 | 0.98 |
| Brunei | 0.26 | 1.18 | −0.40 | −1.60 | −0.22 | −0.82 | 0.73 |

**Table A3.** *Cont.*

| Country | ln(ES) | | ln(EE) | | ln(ESUS) | | R² |
|---|---|---|---|---|---|---|---|
| | Coefficient | t−Value | Coefficient | t−Value | Coefficient | t−Value | |
| Canada | 0.16 | 1.28 | −0.21 | −0.31 | 0.36 ** | 2.60 | 0.99 |
| Chile | 0.02 | 0.14 | 0.20 | 1.42 | 0.31 | 1.70 | 0.99 |
| Cyprus | 0.06 | 1.43 | 1.91 *** | 3.67 | −0.03 | −0.33 | 0.99 |
| Czech Rep | 0.24 | 0.99 | 0.93 | 1.74 | 0.40 * | 1.84 | 0.98 |
| Denmark | 0.22 * | 1.91 | −0.86 | −1.07 | 0.24 *** | 4.01 | 0.98 |
| Estonia | 0.13 | 0.87 | 1.78 *** | 4.27 | −0.10 | −0.71 | 0.98 |
| Finland | −0.31 | −1.32 | −0.60 | −1.10 | −0.02 | −0.19 | 0.94 |
| France | 0.01 | 0.24 | −0.11 | −0.30 | −0.10 | −0.79 | 0.98 |
| Germany | 0.28 ** | 2.26 | −0.48 | −0.78 | −0.13 | −0.92 | 0.97 |
| Greece | 0.03 | 0.16 | −0.03 | −0.07 | 0.54 ** | 2.62 | 0.96 |
| Hungary | −0.08 | −0.47 | 1.36 ** | 2.39 | 0.78 *** | 3.02 | 0.96 |
| Iceland | 1.59 *** | 3.70 | 6.68 * | 1.78 | −0.05 | −0.26 | 0.98 |
| Israel | −0.01 | −0.17 | −0.61 | −1.60 | 0.07 | 1.54 | 0.99 |
| Italy | −0.05 | −1.10 | 0.34 | 1.24 | 0.08 | 0.70 | 0.92 |
| Japan | 0.17 *** | 3.26 | −0.42 | −1.06 | −0.32 ** | −2.95 | 0.97 |
| Korea Rep | −0.27 | −1.48 | 0.24 | 0.36 | 0.15 | 1.24 | 0.99 |
| Latvia | 0.58 | 1.46 | −0.02 | −0.02 | −0.33 | −0.40 | 0.90 |
| Lithuania | 0.10 | 0.29 | 1.42 | 1.17 | −0.22 | −0.38 | 0.93 |
| Luxembourg | 0.09 | 0.88 | −10.29 | −0.84 | −0.44 ** | −2.35 | 0.97 |
| Malta | 0.01 | 0.01 | −0.14 | −0.20 | −0.14 | −0.34 | 0.98 |
| Netherland | −0.19 | −1.30 | 0.19 | 0.11 | 0.19 | 1.36 | 0.96 |
| New Zealand | 0.01 | 0.14 | 0.07 | 0.23 | 0.16 | 1.56 | 0.99 |
| Norway | −0.45 ** | −2.30 | 0.21 | 0.31 | 0.12 | 1.07 | 0.98 |
| Oman | 0.21 | 1.14 | −0.28 | −1.30 | 0.15 | 1.64 | 0.99 |
| Poland | 0.45 | 1.07 | −0.48 | −0.82 | −0.17 | −0.44 | 0.98 |
| Portugal | 0.23 *** | 3.24 | −0.41 | −1.08 | −0.01 | −0.16 | 0.90 |
| Saudi Arabia | 0.11 | 0.45 | −0.18 | −0.78 | −0.03 | −0.26 | 0.99 |
| Slovak Rep | 0.63 | 1.42 | −2.14 | −1.09 | 1.07 | 1.41 | 0.94 |
| Slovenia | 1.49 ** | 2.95 | 0.04 | 0.08 | 0.43 | 1.48 | 0.96 |
| Spain | 0.02 | 0.15 | −0.68 | −1.38 | 0.43 * | 1.79 | 0.96 |
| Sweden | −0.65 | −1.45 | 0.25 | 0.04 | 0.19 | 1.25 | 0.98 |
| Switzerland | −0.11 | −0.57 | −2.19 | −1.41 | 0.08 | 0.39 | 0.99 |
| United Arab Emirates | −1.33 *** | −5.11 | 0.91 ** | 2.37 | −0.47 ** | −2.15 | 0.98 |
| United Kingdom | 0.03 | 0.23 | 2.12 | 1.76 | −0.18 | −0.79 | 0.98 |
| United States | −0.02 | −0.29 | −0.61 | −0.89 | 0.15 | 1.22 | 0.99 |
| Uruguay | 0.42 *** | 3.65 | 0.05 | 0.42 | 0.35 *** | 3.30 | 0.99 |

Note: *, **, and *** indicate the significant level of 10%, 5%, and 1%, respectively. The numbers in parentheses are t−Values.

## Appendix B

**Table A4.** Country List by Regions.

| Region | | (+) ES | (−) ES | (+) EE | (−) EE | (+) ESUS | (−) ESUS |
|---|---|---|---|---|---|---|---|
| Africa | Northern Africa | Morocco | | Egypt, Morocco | | | |
| | Sub-Saharan Africa | | | | | | |
| | (1) Middle Africa | | | Angola | | | Chad, Angola, Gabon |
| | (2) Eastern Africa | Kenya | Mozambique | Ethiopia, Mozambique, Tanzania | | Kenya | Madagascar |
| | (3) Western Africa | Niger | | Côte d'Ivoire | | Nigeria | Niger |
| | (4) Southern Africa | South Africa | | Eswatini, South Africa | | Eswatini | |
| Number of country (% of full sample country) | | 4 (3.67%) | 1 (0.92%) | 9 (8.26%) | 0 | 3 (2.75%) | 5 (4.59%) |
| Asia | Central Asia | | | Kazakhstan | | | |
| | Eastern Asia | China, Japan | | Mongolia, China | | | Japan |
| | South-eastern Asia | Cambodia, Malaysia | | Indonesia, Vietnam | Cambodia | | Indonesia |
| | Southern Asia | Bangladesh | | Iran, Sri Lanka | | | Nepal |
| | Western Asia | Jordan, Bahrain | United Arab Emirates | Armenia, Lebanon Bahrain, Cyprus United Arab Emirates | | Jordan | United Arab Emirates |
| Number of country (% of full sample country) | | 7 (6.42%) | 1 (0.92%) | 12 (11.01%) | 1 (0.92%) | 1 (0.92%) | 4 (3.67%) |
| Americas | Caribbean | | | | | Dominican, Republic | |
| | Central America | Nicaragua, Guatemala | Honduras, Costa Rica | Mexico | | El Salvador, Guatemala | Nicaragua |
| | Northern America | | | | | Canada | |
| | South America | Brazil, Uruguay | | Peru | | Peru, Uruguay | Argentina, Ecuador |
| Number of country (% of full sample country) | | 4 (3.67%) | 2 (1.83%) | 2 (1.83%) | 0 | 6 (5.5%) | 3 (2.75%) |
| Europe | Eastern Europe | Romania | Austria | Ukraine, Romania, Russia, Hungary | | Czech, Hungary | |
| | Northern Europe | Denmark, Iceland | Norway | Estonia, Iceland | | Denmark | |
| | Southern Europe | Albania, Montenegro Portugal, Slovenia | Bosnia and Herzegovina | Albania | | Greece, Spain | Bosnia and Herzegovina |
| | Western Europe | Germany | | | Belgium | | Luxembourg |
| Number of country (% of full sample country) | | 8 (7.34%) | 3 (2.75%) | 7 (6.42) | 1 (0.92%) | 5 (4.59%) | 2 (1.83%) |

Note: This table is modified by Appendix A for regions. Countries by regions are based on the United Nations (UN) criterions (https://unstats.un.org/unsd/methodology/m49/ (accessed on 20 January 2022)).

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
