# Peer review of "An Analysis of the Relationship between Energy Trilemma and Economic Growth"

_sustainability, doi:10.3390/su14073863_

Round 1
Reviewer 1 Report
Authors wisely analyzed the relation between energy trilemma and
economic growth. According to me, the article seems ok for publication. However, there is always scope for improvements.
Reviewer 2 Report
This study analyzed the relationship between energy trilemma (ET) and economic growth in 109 countries between 2000 and 2020 across income levels and regions. This topic is meaningful and the material of this study is relatively comprehensive, but in many aspects, it needs to be carefully revised.
- The abstract part should focus on an overview of the article, while the methodological differences between the method used in this article and previous articles should be explained in the introduction section.
- The abstract section should give highlights of the conclusion of the article, not a general conclusion.
- L98-L99: Who are “the authors” referring to here?
- This paper summarizes the previous research, but does not address the shortcomings of the previous research, or what is the significance of the research in this paper compared with the previous research? What is the purpose of analyzing the relationship between ET and economic growth by income level?
- L171-L183: Both paragraphs describe “the top WETI performers in 2020”, it is suggested that they can be merged.
- L184: Change “if” to “of”.
- L186: Can the name of the table be more complete?
- L208: Are the images stretched? I feel like the fonts are a little distorted.
- What is the difference between this study and many previous studies? Especially in the conclusions? It is recommended that the authors compare the conclusions of this article with those of previous articles.
- L420 and L431: "Second" is used in two places, can it be replaced?
- L305: Change "removes" to "remove”.
Reviewer 3 Report
Overall good work. A spell check may be good.
Fig 1 should be redrawn for clarity and high quality. As for table Table 2, more explanation is required on the criteria used for the classifications and the improvement calculation.
